# Influence of Chêneau-Brace Therapy on Lumbar and Thoracic Spine and Its Interdependency with Cervical Spine Alignment in Patients with Adolescent Idiopathic Scoliosis (AIS)

**DOI:** 10.3390/jcm10091849

**Published:** 2021-04-23

**Authors:** Wojciech Pepke, Aly El Zeneiny, Haidara Almansour, Thomas Bruckner, Stefan Hemmer, Michael Akbar

**Affiliations:** 1Clinic for Orthopedics and Trauma Surgery, Heidelberg University Hospital, 69118 Heidelberg, Germany; wojciech.pepke@med.uni-heidelberg.de (W.P.); elzeneiny@gmail.com (A.E.Z.); Stefan.Hemmer@med.uni-heidelberg.de (S.H.); 2Department of Diagnostic and Interventional Radiology, Eberhard-Karls University, 72076 Tuebingen, Germany; haidara.al-mansour@med.uni-tuebingen.de; 3Institute of Medical Biometry and Informatics, University of Heidelberg, 69120 Heidelberg, Germany; bruckner@imbi.uni-heidelberg.de; 4Clinic for Spine Diseases and Therapies, Meoclinic, 10117 Berlin, Germany

**Keywords:** adolescent idiopathic scoliosis, Chêneau, brace, Cobb angle, sagittal alignment, cervical spine, axial rotation

## Abstract

Chêneau-brace is an effective therapy tool for treatment in adolescent idiopathic scoliosis (AIS). Data on potential interdependent changes of the sagittal profile including the cervical spine are still sparse. The purpose of this study was to evaluate in-brace changes of the thoracic and lumbar spine and their influence on the pelvis and the cervical spine and apical vertebral rotation was reported. Ninety-three patients with AIS undergoing Chêneau-bracing were included. Patients were stratified by lumbar, thoracic and global spine alignment into normolordotic vs. hyperlordotic or normokyphotic vs. hypokyphotic or anteriorly aligned vs. posteriorly aligned groups. The coronal Cobb angle was significantly decreased in all groups indicating good correction while in-brace therapy. Sagittally, in-brace treatment led to significant flattening of lumbar lordosis (LL) in all stratified groups. Thoracic kyphosis (TK) was significantly flattened in the normokyphotic group, but no TK changes were noticed in the hypokyphotic group. Pelvic tilt (PT) stayed unchanged during the in-brace therapy. Chêneau-brace showed marginal changes in the lower cervical spine but had no influence on the upper cervical spine. The apical vertebral axis in primary and secondary curves was unchanged during the first radiological follow-up. Results from this study contribute to better understanding of initial spine behavior in sagittal and axial plane in the context of bracing.

## 1. Introduction

Adolescent idiopathic scoliosis (AIS) is a deformity of the spine including deviation of a coronal curve, axial vertebral rotation, and flattening of the sagittal profile [1,2].

Treatment modalities of AIS range from conservative therapy with physiotherapy and sports, brace therapy, to surgery [3]. Brace treatment is an important player in non-operative treatment in adolescents who are still growing (Risser grade 0–2) with structural Cobb curves >20°, but before surgical threshold is reached [4,5]. The aim of the brace therapy is to guide the spine or at least to stop the curve progression during the pubertal growth spurt [6]. The idea of Chêneau-brace construction contains the use of pressure areas and expansion chambers for three-dimensional treating of scoliosis [7]. The changes of the thoracic and lumbar spine as well as the pelvis during brace therapy were the focus of many previous investigations [1,2]. However, marginal efforts have been put forth to understand the interdependency of cervical spine alignment with regional parameters. Previous studies revealed the interdependency of cervical lordosis (CL), thoracic kyphosis (TK) and global spine posture [8,9,10,11,12,13]. However, there is still lack of information about interaction between spinopelvic and global posture parameters with cervical alignment during the brace therapy.

Within past few years, for the first time the influence of the brace therapy on sagittal profile and transverse plane of the spine of AIS patients was reported [14,15]. These findings show that bracing can have limited corrective effects on the scoliotic curves in sagittal and axial plane [14]. It was revealed that correction of thoracic kyphosis and axial de-rotation is often difficult to achieve [14]. It is still a controversial issue if Chêneau-brace therapy leads to a significant de-rotation in the axial plane which reveals the lack of understanding of 3D-correction of the spine in AIS. Several factors account for planning brace therapy: the complex shape of the deformity, the difficulty of planning, and implementation of an effective correction strategy. Compensating mechanisms are usually not accounted for when planning a brace fabrication. The importance of paying attention to sagittal parameters prior, and during brace treatment could be verified by Matsumoto et al. This group could show a higher risk for curve progression during brace therapy when sagittal profile was not addressed enough with orthosis [16].

The purpose of this study was two-fold: Firstly, to investigate in-brace alteration of the regional and global spinopelvic parameters and the reciprocal changes in cervical alignment. Secondly, to examine brace efficiency according to axial de-rotation.

## 2. Materials and Methods

### 2.1. Study Cohort

This is a retrospective single-center study of AIS patients with a Cobb angle greater than 20° and less than 40°, in whom therapy began between 2010 and 2019. AIS Patients were included with Risser grade 0–2, who had a first Chêneau-brace prescription, and were in adolescent age [17]. Further inclusion criterium was availability of full-spine radiographs in anterior-posterior and lateral view. Exclusion criteria were patients with congenital, neuromuscular, and syndromic scoliosis. Patients without visible full-spine on radiographs were excluded. Figure 1 illustrates the inclusion process. This study was approved by the ethics committee of Heidelberg University (permission No. S-438/2019).

### 2.2. Data Collection and Analysis

Radiographs were performed using conventional radiography or stereoradiography (EOS-imaging^®^, Paris, France) for imaging of the full spine before and during the first in-brace radiographic controls. First in-brace radiographies were performed after completed manufacturing of the Chêneau-brace. Standard duration of brace production lasted from two to three months. Patients were barefoot, had both upper extremities crossed over the chest, and were instructed to look straight ahead in a relaxed position. Radiographs that failed to fulfill these requirements were excluded. Radiography data was saved as a digital imaging and communications in medicine file (DICOM) and exported. Analysis of the radiographs was performed with validated software (Surgimap^®^, New York, NY, USA) [18] by a single reader (AEZ), a research fellow with a medical background. Radiographic parameters included the following:

Coronal parameters: Cobb angle for main curve (MC_COBB), secondary curve (SC_COBB) and tertiary curve (TC_COBB); apex deviation for primary curve (MC_ApexDev), secondary curve (SC_ApexDev) and tertiary curve (TC_ApexDev), coronal alignment (Calignment), C7-plumbline (C7PL).

Sagittal spinopelvic parameters: C0C1 angle (C0C1), C1C2 angle (C1C2), T1 Slope, T1-CL mismatch (T1-CL), C2 Slope, C2-C7 cervical lordosis (CL), T1-T12 thoracic kyphosis TK (T1-T12), T4-T12 thoracic kyphosis TK (T4-T12), T2-T5 angle (T2T5), T5-T12 angle (T5T12), thoraco-lumbar alignment (TL), L1-S1 lumbar lordosis (LL), L1-L4 angle (L1L4), L4-S1 angle (L4S1), pelvic incidence (PI), pelvic tilt (PT), sacral slope (SS), C7-S1 sagittal vertical axis (SVAC7S1), cervical C2-C7 sagittal vertical axis (cSVAC2C7), T1-spinopelvic inclination (T1 SPi), T9-spinopelvic inclination (T9 SPi), T1-pelvic angle (TPA), C2-pelvic angle (CPA), and cervical-thoracic pelvic angle (CTPA).

Axial plane parameters: apical vertebral rotation (AVR) of the primary curve (Raimondi 1) and the secondary curve (Raimondi 2). Raimondi rotation angle is a reliable method for estimating vertebral rotation as projected on standard radiographs of the spine in standing position [19].

### 2.3. Patient Stratification

According to normative values of sagittal profile in children and in adolescents, as published by Mac-Thiong et al. [20], our study group was stratified by their lumbar alignment into hyperlordotic (>59.7°) or normolordotic (36.3° to 59.7°) and their thoracic alignment (T1T12) into hypokyphotic (<33.1°) or kyphotic (33.1° to 54.9°). Patients were considered to have an anterior alignment if SVA was >0 mm and a posterior alignment if SVA was ≤0 mm. This stratification allows the investigation of LL, TK, and SVA variance, its potential changes while brace therapy, and emphasizes the differences between the subgroups.

### 2.4. Statistical Analysis

Software package SPSS^®^ Version 25 (IBM^®^, Armonk, NY, USA) was used for statistical analysis (Figure 2). Data is portrayed as mean and standard deviation. Intergroup comparisons were conducted using paired t-test. Threshold for statistical significance was set at *p* < 0.05. Absolute values of the coronal plane and axial vertebral rotation were utilized for recognition of the severity of rotation, but without considering the direction of rotation. For CL, TK, LL and all segmental angles in sagittal plane, negative values denote lordosis and positive values denote kyphosis.

## 3. Results

### 3.1. Global Analysis

A total of 93 patients (75.3% females, *n* = 70) with a mean age 13.3 ± 2.5 years were included in the study. 38.7% (*n* = 36) of patients had a Risser grade 0, 18.3% (*n* = 17)—Risser grade 1 and 43% (*n* = 40)—Risser grade 2.

In terms of main and secondary scoliotic curve, significant correction of the Cobb angle could be achieved (MC_COBB pre vs. MC_COBB in: 30.2° ± 8.0 vs. 19.8° ± 9.4, *p* < 0.001) (SC_COBB pre vs. SC_COBB in: 22.9° ± 6.7 vs. 19.0° ± 7.7, *p* < 0.001) while brace therapy. Apex deviation of the main curve was also significantly decreased (MC_ApexDev pre vs. MC_ApexDev in 25.3 mm ± 11.1 vs. 13.3 mm ± 9.7, *p* < 0.001). C7-PL and coronal alignment were unchanged (Figure 3).

In terms of sagittal alignment, in-brace patients underwent a loss of LL (LL pre vs. LL in: −54.3° ± 17.2 vs. −48.0° ± 20.1, *p =* 0.011) mainly in the upper lumbar part (L1L4 pre vs. L1L4 in: −24.7° ± 10.5 vs. −19.3° ± 12.7, *p <* 0.001). Partial changes of the pelvic parameters were noted (PI-LL pre vs. PI-LL in: −5.9° ± 12.5 vs. −1.9° ± 12.1, *p <* 0.001) (SS pre vs. SS in: 41.1° ± 10 vs. 38.8° ± 10, *p =* 0.002). PT as a parameter for the compensatory mechanism of the pelvis was unchanged. Due to changes of LL, TK and segmental kyphosis of the upper thoracic spine were also significantly decreased (TK (T1-T12) pre vs. TK (T1-T12) in: 32.8° ± 14.0 vs. 26.4° ± 13.3, *p <* 0.001) (TK (T4-T12) pre vs. TK (T4-T12) in: 26.2° ± 12.3 vs. 23.2° ± 10.6, *p <* 0.001) (T2T5 pre vs. T2T5 in: 13.5° ± 8.2 vs. 11.1° ± 7.4, *p =* 0.004) (T5T12 pre vs. T5T12 in: 21.5° ± 11.4 vs. 19.2° ± 9.4, *p =* 0.002). Furthermore, in-brace patients had a significant decrease of T1 Slope (T1 Slope pre vs. T1 Slope in: 16.3° ± 9.0 vs. 13.5° ± 8.6, *p =* 0.001). Consequently, T1-CL and C2 Slope were also significantly lower while brace therapy (T1-CL pre vs. T1-CL in: 26.2° ± 11.5 vs. 23.1° ± 11.9, *p =* 0.032) (C2 Slope pre vs. C2 Slope in: 26.8° ± 8.8 vs. 24.1° ± 9.1, *p =* 0.001). Interestingly, in this whole study population, mean CL was kyphotic and remained to be unchanged during the brace treatment (CL pre vs. CL in: 10.7° ± 13.6 vs. 10.5° ± 13.9). Due to a decrease in the T1 Slope, decrease of the C2 Slope without any changes of kyphotic CL, cSVAC2C7 was also more posteriorly aligned (cSVAC2C7 pre vs. cSVAC2C7 in: 25.0 mm ± 9.9 vs. 22.1 mm ± 8.2, *p =* 0.003). In this study population, the parameters of upper cervical spine were unchanged (Figure 4 and Figure 5) (Table 1).

No statistically different changes of AVR for main (Raimondi 1) and secondary (Raimondi 2) curves could be observed (Figure 3).

### 3.2. Stratification by LL

In the lumbar normolordotic group, brace therapy modulated the lumbar shape without significant changing of LL. In-brace patients have a decreased L1L4 angle (L1L4 pre vs. L1L4 in: −20.1° ± 9.4 vs. −16.8° ± 8.9, *p =* 0.009), but slightly increased L4S1 without reaching a level of significance. Except PI-LL and SS, no other pelvic parameters were changed (PI-LL pre vs. PI-LL in: −4.1° ± 11.9 vs. −0.7° ± 12.0, *p =* 0.004) (SS pre vs. SS in: 36.5° ± 7.6 vs. 34.7° ± 8.8, *p =* 0.046). In this study group, also TK was significantly flattened while brace therapy (TK (T1T12) pre vs. TK (T1T12) in: 29.9° ± 11.4 vs. 24.9° ± 11.8, *p =* 0.001) (TK (T4T12) pre vs. TK (T4T12) in: 23.2° ± 11.3 vs. 21.0° ± 9.5, *p =* 0.018). Consequently, also T1-Slope and C2-Slope were significantly diminished (T1-Slope pre vs. T1-Slope in: 16.2° ± 8.6 vs. 12.8° ± 8.5, *p =* 0.004) (C2-Slope pre vs. C2-Slope in: 26.9° ± 7.9 vs. 24.7° ± 8.7, *p =* 0.048). Kyphotic CL, cSVAC2CC7 and all parameters of upper cervical spine revealed to be unchanged (Figure 6) (Table 1).

In-brace patients of the hyperlordotic group revealed significant flattening of LL, mainly of the upper lumbar part (LL pre vs. LL in: −67.0° ± 6.8 vs. −52.15° ± 30.3, *p =* 0.007) (L1L4 pre vs. L1L4 in: −32.3° ± 7.3 vs. −23.4° ± 16.5, *p =* 0.004). Due to pelvic parameters, only PI-LL mismatch and SS were changed (PI-LL pre vs. PI-LL in: −8.9° ± 12.9 vs. −3.9° ± 11.9, *p =* 0.006) (SS pre vs. SS in: 48.6° ± 8.9 vs. 45.7° ± 7.7, *p =* 0.015). All measured regional parameters of the thoracic spine were decreased in terms of flattening of TK (TK (T1T12) pre vs. TK (T1T12) in: 37.5° ± 16.3 vs. 28.8° ± 15.4, *p =* 0.002) (TK (T4T12) pre vs. TK (T4T12) in: 31.1° ± 12.4 vs. 26.9° ± 11.4, *p =* 0.002) (T2T5 pre vs. T2T5 in: 15.1° ± 8.0 vs. 12.2° ± 8.6, *p =* 0.029) (T5T12 pre vs. T5T12 in: 25.9° ± 11.4 vs. 21.9° ± 10.0, *p =* 0.004). In comparison to pre-treatment status, in-brace patients were more anteriorly aligned (SVAC7S1 pre vs. SVAC7S1 in: −16.1 mm ± 27.6 vs. −3.8 mm ± 25.5, *p =* 0.048) (T1SPi pre vs. T1SPi in: −5.3° ± 3.2 vs. −3.5° ± 3.5, *p =* 0.028). In this group, T1Slope revealed a slight decrease without reaching the level of significance (*p =* 0.064). Furthermore, C2-Slope and cSVAC2C7 were significantly decreased (C2-Slope pre vs. C2-Slope in: 26.7° ± 10.2 vs. 23.1° ± 9.9, *p =* 0.003) (cSVAC2C7 pre vs. cSVAC2C7 in: 26.7° ± 8.4 vs. 21.6° ± 9.2, *p =* 0.002). In contrast to that, cervical kyphosis (CL), and all parameters of upper cervical spine were unchanged (Figure 6) (Table 1).

### 3.3. Stratification by TK

In the thoracic normokyphotic group, in-brace patients had a significantly diminished TK (T1-T12) (TK (T1-T12) pre vs. TK (T1-T12) in: 42.2 ± 6.2 vs. 32.7 ± 12.0, *p <* 0.001), TK (T4-T12) (TK (T4-T12) pre vs. TK (T4-T12) in: 34.5 ± 8.8 vs. 29.1 ± 10.1, *p <* 0.001). In contrast, the segmental angle of the upper thoracic spine (T2T5 angle) increased significantly (T2T5 angle pre vs. T2T5 angle in: −4.9 ± 3.3 vs. 13.5 ± 6.9, *p <* 0.001). In this group, in-brace patients had a significant decrease of LL (LL pre vs. LL in: −59.2 ± 10.4 vs. −49.4 ± 23.2, *p =* 0.012), caused by loss of lordosis in the upper part of lumbar spine L1L4 (L1L4 pre vs. L1L4 in: −27.2 ± 9.8 vs. −20.8 ± 13.5, *p =* 0.004). Due to pelvic parameters, no significant changes were measured with exception of PI-LL (PI-LL pre vs. PI-LL in: −11.1 ± 11.7 vs. −5.3 ± 12.0, *p <* 0.001). In-brace patients had a significant flattening of T1-Slope (T1-Slope pre vs. T1-Slope in: 21.5 ± 6.6 vs. 17.1 ± 8.4, *p =* 0.001) and had more posteriorly aligned cervical spine (cSVAC2C7 pre vs. cSVAC2C7 in: 26.7 ± 9.7 vs. 23.3 vs. 8.9, *p =* 0.010). Furthermore, slight progress of cervical kyphosis (CL) without reaching a level of significance was noticed (*p =* 0.077). Finally, in this group the parameters of the upper cervical spine were unchanged (Figure 7) (Table 1).

In-brace patients of the thoracic hypokyphotic group showed no significant changes of the thoracic spine. However, significant flattening of the upper lumbar part could be measured (L1L4 pre vs. L1L4 in: −21.9 ± 9.9 vs. −18.6 ± 9.5, *p =* 0.015) without alteration of LL (−49.2 ± 20.0 vs. −49.4 ± 23.3, *p =* 0.639). All pelvic parameters were unchanged. Due to cervical parameters, with the exception of the significantly decreased C2-Slope (C2-Slope pre vs. C2-Slope in: 27.6 ± 8.2 vs. 23.6 ± 8.3, *p =* 0.001), no other parameters of lower and upper cervical spine were changed (Figure 7) (Table 1).

### 3.4. Stratification by SVA

In-brace patients of the anteriorly aligned group revealed significant flattening of the TK (TK (T1-T12) pre vs. TK (T1-T12) in: 29.7 *±* 12.7 vs. 25.3 *±* 11.5, *p =* 0.019) (TK (T4-T12) pre vs. TK (T4-T12) in: 23.6 *±* 12.5 vs. 20.6 *±* 10.0, *p =* 0.023) and flattening of the upper lumbar spine (L1L4 pre vs. L1L4 in: −25.0 *±* 11.1 vs. −18.7 *±* 14.2, *p =* 0.04). Due to pelvic parameters, there was a significant decrease of SS (SS pre vs. SS in: 43.8 *±* 10.6 vs. 40.6 *±* 10.8, *p =* 0.008) and increase of PT (PT pre vs. PT in: 11.8 *±* 8.1 vs. 14.9 *±* 9.1, *p =* 0.003). Brace therapy of this group led to a significant posterior shift of SVAC7S1 (SVAC7S1 pre vs. SVAC7S1 in: 21.2 *±* 21.6 vs. 6.5 *±* 30.8, *p =* 0.014) and posterior increase of T1Spi (T1Spi pre vs. T1Spi in: −1.4 *±* 2.5 vs. −3.3 *±* 3.2, *p =* 0.06). Cervical parameters where unchanged with the exception of decrease in the T1-Slope (T1-Slope pre vs. T1-Slope in: 17.8 *±* 8.7 vs. 13.9 *±* 8.8, *p =* 0.019) and the C2-Slope (C2-Slope pre vs. C2-Slope in: 28.4 *±* 8.6 vs. 25.2 *±* 9.1, *p =* 0.015). Cervical kyphosis (CL) and the parameters of the upper cervical spine were unchanged (Figure 8) (Table 1).

The posteriorly aligned group suffered while in-brace therapy a highly significant flattening of all thoracic parameters (TK (T1-T12) pre vs. TK (T1-T12) in: 34.0 *±* 14.3 vs. 26.8 *±* 14.1, *p =* 0.006) (TK (T4-12) vs. TK (T4-T12) in: 27.3 *±* 12.1 vs. 24.3 *±* 10.7, *p =* 0.002) (T2T5 pre vs. T2T5 in: 13.9 *±* 8.1 vs. 11.2 *±* 7.1, *p <* 0.001) (T5T12 pre vs. T5T12 in: 22.6 *±* 11.0 vs. 20.2 *±* 9.5, *p =* 0.009). Simultaneously, the lumbar shape was flattened in the upper lumbar part (L1L4 pre vs. L1L4 in: −24.6 *±* 10.3 vs. −19.6 vs. 12.1, *p =* 0.001) and consequently LL was decreased (LL pre vs. LL in: −55.7 *±* 15.8 vs. −49.2 *±* 17.6, *p =* 0.007). PI-LL mismatch and SS were significantly changed (PI-LL pre vs. PI-LL in: −9.5 *±* 11.2 vs. −4.8 *±* 11.0, *p <* 0.001) (SS pre vs. SS in: 39.9 *±* 9.5 vs. 38.1 *±* 9.5, *p =* 0.042). Interestingly, in-brace patients were significantly more anteriorly aligned (SVAC7S1 pre vs. SVAC7S1 in: −22.6 *±* 17.1 vs. −9.9 *±* 28.5, *p =* 0.002), and their cervical spine—more posteriorly aligned (cSVAC2C7 pre vs. cSVAC2C7 in: 25.0 *±* 9.0 vs. 22.5 *±* 7.9, *p =* 0.021). Analyzing the parameters of the cervical spine, there was a significant decrease of the T1-Slope, T1-CL and C2-Slope (T1-Slope pre vs. T1-Slope in: 15.7 *±* 9.1 vs. 13.3 *±* 8.6, *p =* 0.027) (T1-CL pre vs. T1-CL in: 25.5 *±* 10.7 vs. 21.7 *±* 11.8, *p =* 0.036) (C2-Slope pre vs. C2-Slope in: 26.2 *±* 8.9 vs. 23.6 *±* 9.2, *p =* 0.015). In contrast, cervical kyphosis (CL) and all parameters of the upper cervical spine were unchanged (Figure 8) (Table 1).

## 4. Discussion

This study investigated the influence of bracing on the spine in terms of the coronal, sagittal, and axial changes of the thoracic and lumbar spine as well as on the cervical spine.

The global analysis of this study population revealed a significant Cobb angle correction of the primary and secondary curves, which could be considered an indicator for therapy success [21,22]. Furthermore, a significant decrease of apex deviation was noticed at least for primary curves, which is in line with the coronal Cobb angle correction. In addition, no coronal deviation of the spine occurred during brace therapy. Cobb correction revealed similar results as correction mentioned in the literature [23,24]. Although there exists a lot of high-quality studies on the efficacy of bracing [25,26,27], there is still a need for additional evidence on fundamental topics, such as impact of the brace therapy on sagittal alignment and axial derotation.

Previous studies of brace efficacy focused on coronal curves, which included only a few parameters relating to the sagittal profile such as TK and LL [28,29,30]. However, patients in these studies were not stratified by the magnitude of TK or LL. In the present cohort, sagittal alignment was evaluated and compared with the normative values of children and adolescents [20]. Therefore, this stratification sheds light on changes of the sagittal profile through brace therapy in terms of LL, TK, C7S1SVA, and their possible impact on cervical alignment. This stratification has already been used for an in-depth analysis of the sagittal profile in AIS patients [12,31]. The sagittal alignment of the lumbar (especially upper lumbar part, L1L4) spine and all regional parameters of the thoracic spine were significantly flattened during brace therapy. Slight influence on pelvic parameters (PI-LL, SS) without significant use of compensatory mechanisms (PT) was noticed. PT is a powerful parameter that is expressed if pelvic compensatory mechanisms are used [32,33]. Patients underwent no significant alteration of PT indicating no relevant compensatory mechanisms of the pelvis to preserve a straight position of the trunk. This phenomenon might be explained by in-brace simultaneous flattening the upper lumbar spine (L1L4) and TK, but without significant changes in the lower lumbar part (L4S1), T1SPi and SVAC7S1. Furthermore, the cervical spine revealed a significant decrease of cervical parameters as the T1-Slope and C2-Slope, but without any changes of CL. Unchanged CL may be explained by the kyphotic cervical alignment of this study cohort, which might implicate fewer compensation possibilities, also in young patients. It was convenient to decrease the T1-Slope and C2-Slope without changes of CL, and cSVAC2C7 was also more posteriorly aligned and changed, reaching the level of significance. Therefore, we postulate that brace therapy influences the posture of the lower cervical spine in a narrow range and does not lead to relevant alignment changes. Hence, no significant changes in the upper cervical parameters could be measured. Therefore, brace therapy did not have any influence on the upper cervical spine in preserving the horizontal gaze.

Moreover, due to LL-stratification-group, lordosis of the upper lumbar spine (L1L4) was flattened significantly in both groups (normolordotic and hyperlordotic group) and caused significant reduction of LL (L1S1) during brace therapy. This fact indicates the potential influence of brace treatment on lumbar spine alignment in all patients who underwent brace treatment. Interestingly, brace therapy revealed significant flattening of TK in normokyphotic, but not in hypokyphotic patients, which might be explained by anatomical conditions of the trunk and brace construction principles. In this stratified cohort there was still an outstanding reaction of flattening of LL in both the normo- and hypokyphotic group. In addition, these groups revealed significant changes of PI-LL mismatch without alteration of PT. Similar changes in terms of flattening of LL and TK was found in both SVAC7S1-stratified groups. Moreover, all groups revealed changes of PI-LL mismatch due to LL flattening. Nevertheless, except for the anteriorly aligned group due to SVA stratification, all other groups revealed no change of PT. In all groups, during brace therapy, PT was in normal range (less than 20°), indicating no need for pelvic compensatory mechanisms due to alteration of the trunk. A significant increase of PT in anteriorly aligned group could be a result of a significant decrease of SVAC7S1 without LL flattening in this group. Additionally, in all these stratified groups there was minimal influence on cervical posture without significant alignment changes of the lower cervical spine. In all groups, the upper cervical spine parameters remained to be unchanged. Therefore, brace therapy did not have any influence on the upper cervical spine in all stratified groups as well.

AVR correction during brace treatment still remains a challenging therapy target [15,28,34,35]. In this study, no significant AVR correction could be measured for primary and secondary curves. This phenomenon may be explained by several factors. 3D full-spine analysis of teleradiographs of AIS patients revealed significant detorsion of in-brace patients with structural curves in the thoracolumbar part, but not in the thoracic or lumbar curves [15]. Therefore, the primary curves of thoracolumbar junction seem to be easier to correct AVR with brace therapy [15]. This in turn might be explained by construction principles and three-point fulcrum concept of Chêneau-braces. In this study, the stratification was chosen for the exploration of possible brace influence on sagittal profile of the spine and hence our study cohort was not stratified among coronal curve topography in the thoracic, thoracolumbar, or lumbar curves. Another important point is the period of brace therapy and its possible influence on AVR correction over a longer therapy time. In this study, we focused on the immediate brace impact on spine alignment. Immediate first in-brace radiographs were evaluated in terms of correction magnitude. It is possible, that this first radiological documentation of correction magnitude does not reveal significant axial correction. On the other hand, long-term brace therapy might have a positive influence on AVR also in this study cohort. Nevertheless, data about axial curve correction in AIS patients is still sparse and this crucial issue requires further research.

One of the limitations of this study group is retrospective study design and the absence of a control group. However, ethical considerations about unnecessary radiographs in normal young volunteers or non-initiation of brace therapy in patients with indication of brace therapy were prohibitive. Moreover, stratification based on Lenke classification [36] was not possible due to resultingly very small subgroups. Furthermore, in-brace correction data cannot directly imply an evaluation of long-term efficacy [28]. Most patients included in this study are still undergoing brace treatment. Therefore, we did not perform a follow-up evaluation. Furthermore, it has been shown in normal volunteers during the growth spurt that the lumbar and thoracic spine underwent changes in the sagittal profile [37,38,39]. This fact indicates potential changes of sagittal profile also in AIS patients, independent of brace therapy. Therefore, a follow-up study after brace treatment with a larger cohort and with comparison to normative values should be performed. Furthermore, it is important to note, that the concomitant rib cage deformity in AIS patients might have a possible negative influence on brace-treatment effectiveness [40,41]. In this study, this factor was not evaluated. Finally, measurement error of the software should be considered. Though, in this study, the software used is validated and a high reliability level of the computerized measurements proved [18].

## 5. Conclusions

Cobb angle of primary and secondary curves was significantly decreased during brace therapy, indicating a good in-brace coronal correction. Sagittally, the impact of the brace lead to loss of LL in all AIS patients of this study cohort. Loss of TK was noticed only in normokyphotic patients and was unchanged in hypokyphotic patients. No pelvic compensation was needed during brace treatment. In-brace patients revealed small changes of the lower cervical parameters such as T1-Slope and C2-Slope, but without alteration of kyphotic CL. The upper cervical spine did not reveal any changes. Therefore, the influence of brace therapy on the lower cervical spine is marginal and not existent on the upper cervical spine. Finally, first in-brace radiographs revealed that no AVR correction ensued. Results from this study could shed some light on spinal behavior in the context of bracing and may be beneficial for treating physicians.

## Figures and Tables

**Figure 1 jcm-10-01849-f001:**
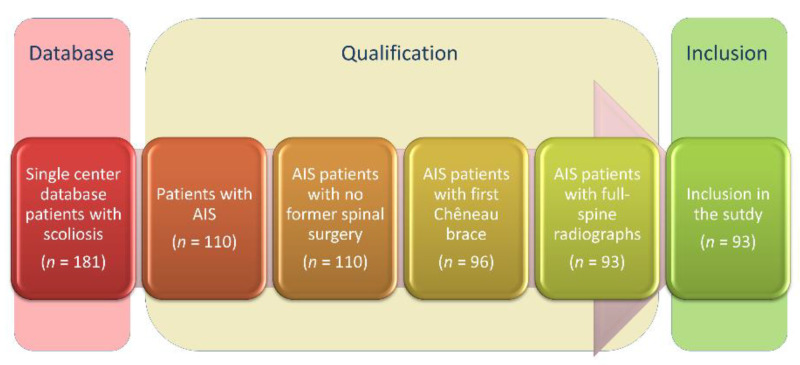
Flow diagram describing the inclusion process of the study population.

**Figure 2 jcm-10-01849-f002:**
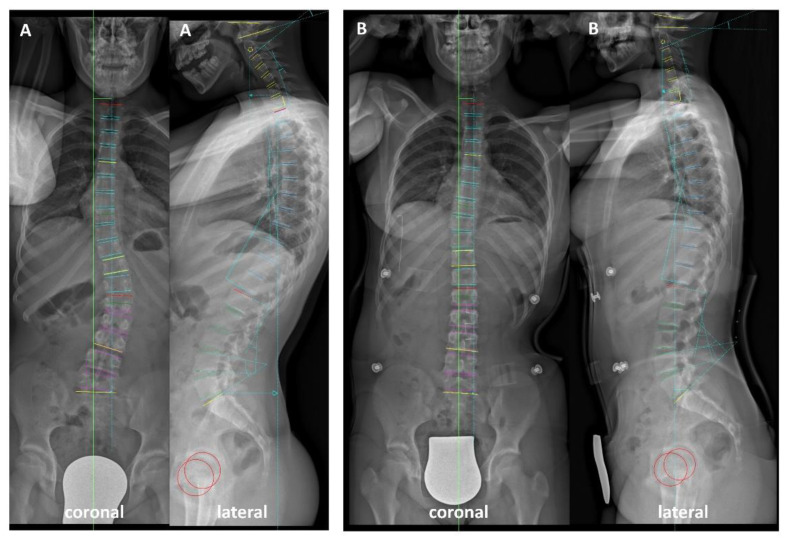
(**A**) Pre- and (**B**) in-brace coronal and lateral radiographs illustrating the analyzed spinal parameters.

**Figure 3 jcm-10-01849-f003:**
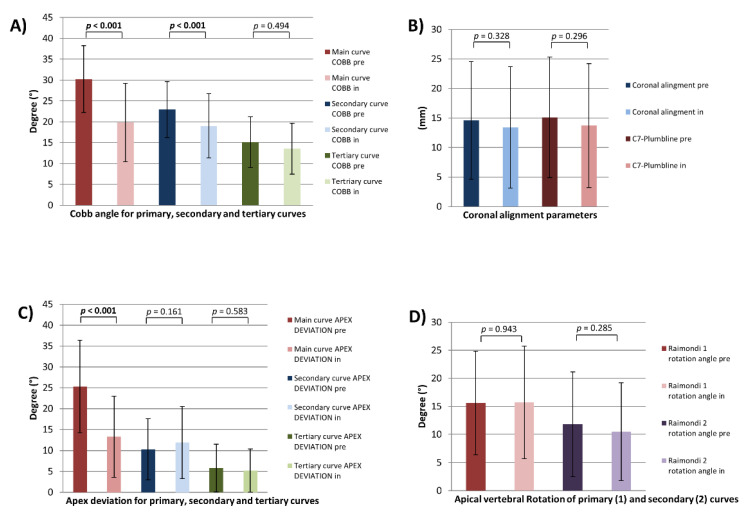
Coronal and axial plane parameters. Cobb angle (**A**), coronal alignment parameters (**B**), apex deviation (**C**) and apical vertebral rotation (AVR) (**D**) for all patients pre- and in-brace. Vertebral rotation was estimated using the Raimondi method. Raimondi 1 angle: AVR of primary curve, Raimondi 2 angle: AVR of secondary curve; *p* = statistical significance, *p* < 0.05.

**Figure 4 jcm-10-01849-f004:**
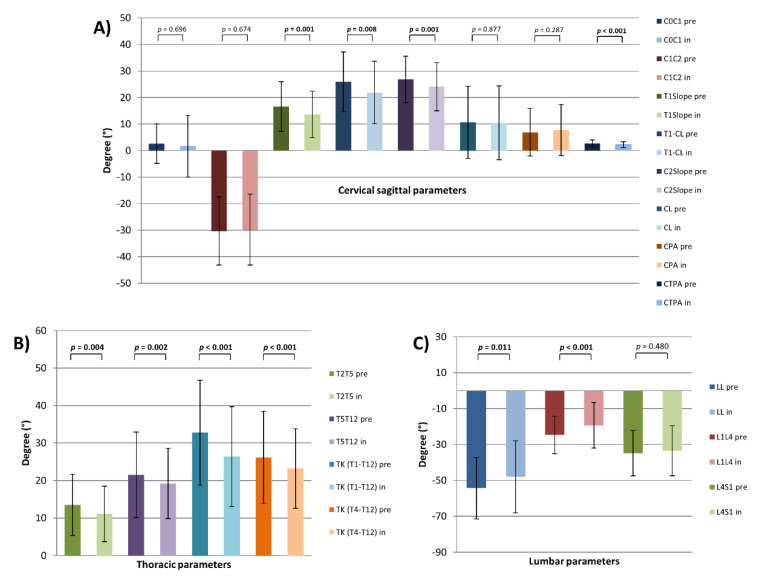
Sagittal plane parameters of cervical (**A**), thoracic (**B**) and lumbar (**C**) spine for all patients (pre- and in-brace); *p* = statistical significance, *p <* 0.05.

**Figure 5 jcm-10-01849-f005:**
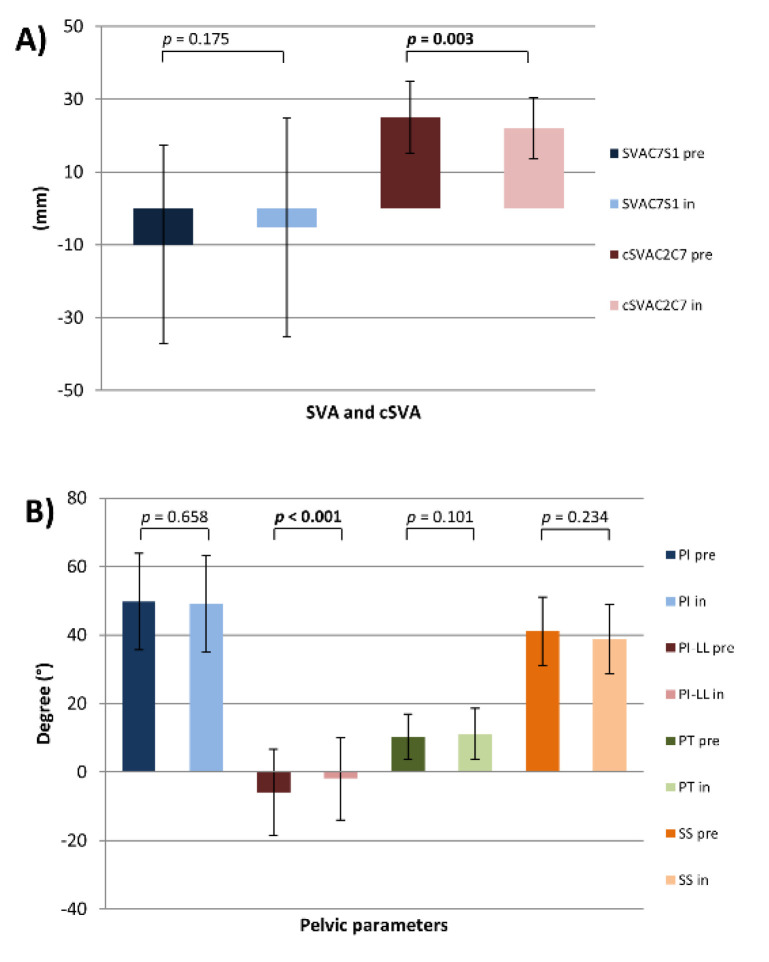
Sagittal plane parameters. (**A**) SVA, cSVA and (**B**) pelvic parameters for all patients (pre- and in-brace); *p* = statistical significance, *p <* 0.05.

**Figure 6 jcm-10-01849-f006:**
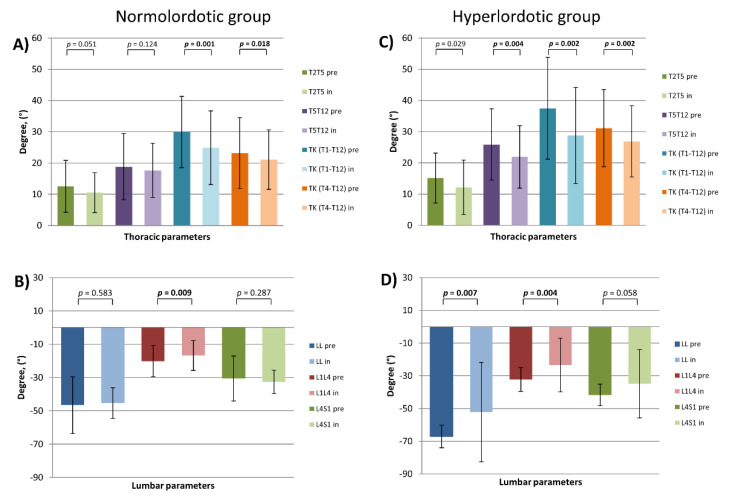
Sagittal thoracic and lumbar parameters by lumbar lordosis (LL) stratified patients (pre- and in-brace). (**A**) thoracic and (**B**) lumbar parameters of the normolordotic group. (**C**) thoracic and (**D**) lumbar parameters of the hyperlordotic group; *p* = statistical significance, *p <* 0.05.

**Figure 7 jcm-10-01849-f007:**
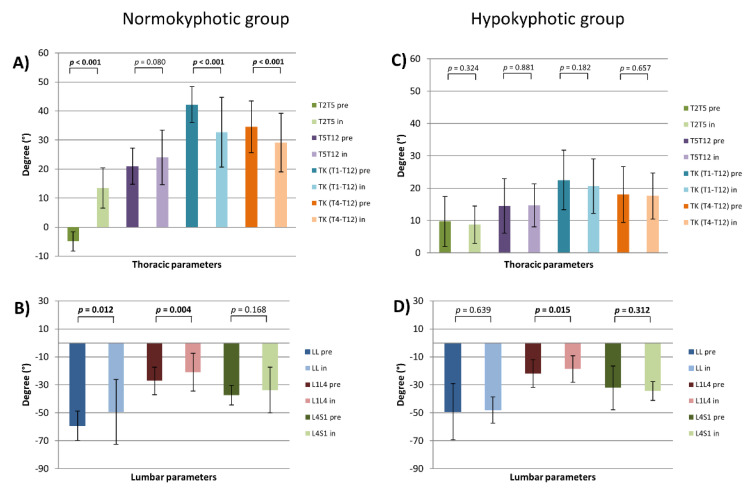
Thoracic and lumbar parameters by thoracic kyphosis (TK) stratified patients (pre- and in-brace). (**A**) thoracic and (**B**) lumbar parameters of the normokyphotic group. (**C**) thoracic and (**D**) lumbar parameters of the hypokyphotic group; p=statistical significance, *p <* 0.05.

**Figure 8 jcm-10-01849-f008:**
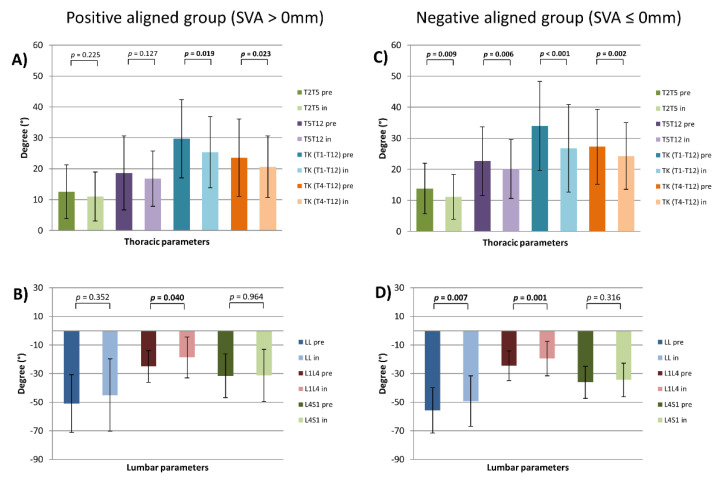
Thoracic and lumbar parameters by sagittal vertical axis (SVA) stratified patients (pre- and in-brace). (**A**) thoracic and (**B**) lumbar parameters of the positive aligned group. (**C**) thoracic and (**D**) lumbar parameters of the negative aligned group; *p* = statistical significance, *p <* 0.05.

**Table 1 jcm-10-01849-t001:** Pre- to in-brace changes of sagittal parameters.

Sagittal Plane	All Patients			Normolordotic Group			Hyperlordotic Group			Normokyphotic Group			Hypokyphotic Group			SVA Positive Group			SVA Negative Group		
Cervical Spine	Mean	SD	*p* value	Mean	SD	*p* value	Mean	SD	*p* value	Mean	SD	*p* value	Mean	SD	*p* value	Mean	SD	*p* value	Mean	SD	*p* value
C0C1 pre	2.6	7.4	0.696	2.8	7.2	0.303	2.3	7.7	0.155	3.8	6.8	0.286	2.2	7.9	0.714	2.9	7.4	0.920	2.6	7.6	0.706
C0C1 in	1.7	11.6		3.9	11		−1.7	12		1.1	14		2.4	11		2.8	7.4		2	12.3	
C1C2 pre	−30.3	12.9	0.674	−28.8	15	0.486	−32.7	9.8	0.206	−29	8.8	0.812	−30.6	16	0.571	−31.3	16.5	0.970	−29.9	11.3	0.577
C1C2 in	−29.8	13.4		−30.6	8.3		−28.3	19		−28.4	13		−32.3	8.4		−31.4	13.6		−28.8	13.5	
C2Slope pre	26.8	8.8	0.001	26.9	7.9	0.048	26.7	10	0.003	25.9	9.7	0.796	27.7	8.2	0.001	28.5	8.6	0.015	26.1	8.9	0.015
C2Slope in	24.1	9.1		24.7	8.7		23.1	9.9		25.6	10		23.6	8.4		25.2	9.1		23.6	9.2	
CL pre	10.7	13.6	0.877	11.5	13	0.552	9.2	15	0.234	5.9	12	0.077	16	13	0.221	10.4	13.9	0.780	10.7	13.5	0.715
CL in	10.5	13.9		12.5	13		7.1	15		9.1	14		14	11		11.1	12.8		10.2	14.3	
T1Slope pre	16.3	9	0.001	16.2	8.6	0.004	17.2	10	0.064	21.5	6.6	0.001	9.7	7.7	0.324	17.8	8.7	0.019	15.7	9.1	0.027
T1Slope in	13.5	8.6		12.8	8.5		14.9	8.7		17.1	8.4		8.7	5.8		13.9	8.8		13.3	8.6	
T1-CL pre	26.2	11.5	0.032	18.8	11	0.124	25.9	11	0.004	21	6.2	0.080	14.5	8.4	0.881	18.6	12.0	0.127	25.5	10.7	0.036
T1-CL in	23.1	11.9		17.6	8.7		21.9	10		24	9.4		14.7	6.7		16.8	9.0		21.7	11.8	
CPA pre	6.9	8.9	0.287	6.7	9.4	0.033	7.4	8.1	0.881	6	9.3	0.582	7.5	8.8	0.056	12.7	8.5	0.771	4.5	8	0.171
CPA in	7.8	9.6		8.2	9.7		7.1	9.5		6.8	10		8.9	8.7		12.3	11.2		5.9	8.1	
CTPA pre	2.7	1.3	0.000	2.5	1.4	0.112	3.1	1.1	0.000	3.2	1.1	0.000	2.2	1.2	0.239	2.3	1.5	0.247	2.9	1.2	0.000
CTPA in	2.2	1.1		2.3	1		2.2	1.3		2.5	1.1		2	0.8		2.0	1.2		2.3	1	
cSVAC2C7 pre	25	9.9	0.003	24	11	0.221	26.5	8.4	0.002	26.7	9.8	0.010	22.7	9.7	0.228	25.6	12.0	0.060	24.7	9.1	0.052
cSVAC2C7 in	22.1	8.2		22.3	7.8		21.6	9		23.3	8.9		20.9	6.9		21.2	9.3		22.7	7.9	
Thoracic Spine	Mean	SD	*p*	Mean	SD	*p*	Mean	SD	*p*	Mean	SD	*p*	Mean	SD	*p*	Mean	SD	*p*	Mean	SD	*p*
T2T5 pre	13.5	8.2	0.004	12.5	8.2	0.051	15.1	8.0	0.028	−4.9	3.3	0.000	9.7	7.7	0.324	12.6	8.7	0.225	13.8	8.1	0.009
T2T5 in	11.1	7.4		10.5	6.4		12.2	8.6		13.5	6.9		8.7	5.8		11.1	7.9		11.2	7.2	
T5T12 pre	21.5	11.4	0.002	18.8	10.6	0.124	25.9	11.4	0.004	21	6.2	0.080	14.5	8.4	0.881	18.6	11.9	0.127	22.6	11.5	0.006
T5T12 in	19.2	9.4		17.5	8.7		21.9	10		24	9.4		14.6	6.6		16.8	8.9		20.2	9.5	
TK (T1-T12) pre	32.8	14	0.000	29.9	11.4	0.001	37.5	16.3	0.002	42.2	6.2	0.000	22.5	9.2	0.182	29.7	12.6	0.019	34.0	14.3	0.006
TK (T1-T12) in	26.4	13.3		24.9	11.8		28.7	15.4		32.7	12		20.6	8.4		25.3	11.4		26.8	14.1	
TK (T4-T12) pre	26.2	12.3	0.000	23.2	11.3	0.018	31.1	12.4	0.002	34.5	8.8	0.000	18	8.6	0.657	23.5	12.5	0.023	27.2	12.1	0.002
TK (T4-T12) in	23.2	10.6		21	9.5		26.8	11.4		29.1	10.1		17.6	7.1		20.6	9.9		24.3	10.7	
TL pre	−2	9.3	0.042	−0.7	9.4	0.101	−4.1	9.1	0.237	−0.01	9.6	0.704	−3.1	8.9	0.007	−3.3	9.7	0.170	−1.4	9.3	0.139
TL in	−0.5	7.9		0.8	7.7		−2.7	7.6		−0.4	7.7		−0.1	8.2		−0.9	8.8		−0.3	7.5	
Lumbar Spine	Mean	SD	*p*	Mean	SD	*p*	Mean	SD	*p*	Mean	SD	*p*	Mean	SD	*p*	Mean	SD	*p*	Mean	SD	*p*
LL pre	−54.3	17.2	0.011	−46.6	17	0.583	−67	6.9	0.007	−59.3	10.4	0.012	−49.2	20	0.639	−50.9	20.1	0.352	−55.7	15.8	0.007
LL in	−48	20.1		−45.4	9.2		−52.2	30		−49.4	23		−48	9.4		−45.0	25.2		−49.2	17.6	
L1L4 pre	−24.7	10.5	0.000	−20.1	9.4	0.009	−32.3	7.3	0.004	−27.2	9.8	0.004	−21.9	9.9	0.015	−25.0	11.1	0.040	−24.6	10.3	0.001
L1L4 in	−19.3	12.7		−16.8	9		−23.4	17		−20.9	14		−18.6	9.5		−18.7	14.2		−19.6	12.1	
L4S1 pre	−34.8	12.6	0.480	−30.6	14	0.287	−41.7	6.6	0.058	−37.4	7	0.168	−32.2	16	0.312	−31.6	15.3	0.964	−36.1	11.2	0.316
L4S1 in	−33.5	13.9		−32.6	7		−34.9	21		−33.8	16		−34.4	6.8		−31.3	18.2		−34.4	11.7	
Pelvis	Mean	SD	*p*	Mean	SD	*p*	Mean	SD	*p*	Mean	SD	*p*	Mean	SD	*p*	Mean	SD	*p*	Mean	SD	*p*
PI pre	49.7	14.1	0.211	44.6	13	0.817	58.2	12	0.080	48.1	14	0.658	50.5	14	0.065	54.9	15.9	0.513	47.6	12.9	0.088
PI in	49.1	14.1		44.7	13		56.4	13		47.6	14		49.6	14		55.3	14.2		46.6	13.3	
PI-LL pre	−5.9	12.5	0.000	−4.1	12	0.004	−8.9	13	0.006	−11.6	11.7	0.000	−0.7	11	0.081	2.9	11.0	0.194	−9.5	11.2	0.000
PI-LL in	−1.9	12.1		−1	12		−3.9	12		−5.3	12		1.4	12		5.2	11.6		−4.8	11.1	
PT pre	10.3	6.7	0.144	10	6.8	0.426	10.8	6.4	0.194	9.6	6.3	0.101	10.8	7	0.903	11.8	8.1	0.003	9.7	5.9	0.797
PT in	11.1	7.5		15.5	8		11.7	6.4		11.2	8		10.9	7.4		14.9	9.1		9.5	6.2	
SS pre	41.1	10	0.002	36.5	7.6	0.046	48.6	8.9	0.015	39.8	9.5	0.234	41.8	10	0.002	43.8	10.6	0.008	39.9	9.5	0.042
SS in	38.8	10		34.7	8.8		45.7	7.7		38.4	11		38.8	9		40.6	10.8		38.1	9.6	

C0C1 = C0C1 angle, C1C2 = C1C2 angle, CL = C2-C7 cervical lordosis, T1-CL = T1-CL mismatch, CPA = C2-pelvic angle, CTPA = cervical-thoracic pelvic angle, cSVAC2C7 = cervical sagittal vertical axis, T2T5 = T2T5 angle, T5T12 = T5T12 angle, TK (T1-T12) = thoracic kyphosis T1-T12, TK (T4-T12) = thoracic kyphosis T4-T12, TL = thoraco-lumbar alignment, LL = lumbar lordosis, L1L4 = L1L4 angle, L4S1 = L4S1 angle PI = pelvic incidence, PI-LL = PI-LL mismatch, PT = pelvic tilt, SS = sacral slope.

## Data Availability

The data presented in this study are available on request from the corresponding author.

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
