# Peer review of "Influence of Chêneau-Brace Therapy on Lumbar and Thoracic Spine and Its Interdependency with Cervical Spine Alignment in Patients with Adolescent Idiopathic Scoliosis (AIS)"

_jcm, 2021, doi:10.3390/jcm10091849_

Round 1
Reviewer 1 Report
The article is an interesting one and all the subtitles of this submission are well written and with balance in terms of the present literature.
However the reviewers would like to note that in the limitations section of the article it would be good the authors to discuss and comment the absence of the thoracic deformity study of their patents’ sample (rib cage hump – transverse plane deformity of rib cage) and the impact of this deformity changes due to bracing on the AVR, (see research and publications of Prof J. Sevastic).
Author Response
Thank you very much for your kind words. We would also like to thank you for raising such an important query. We studied publications of Professor J. Sevastik carefully. We studied also research by Pasha et al. (“3D spinal and rib cage predictors of brace effectiveness in adolescent idiopathic scoliosis”). Transverse plane deformity of rib cage and its influence on correction magnitude during brace therapy is a very interesting issue and should be further investigated.
In this study, our focus was on sagittal, axial and coronal plane of the spine during brace treatment. Stratifying sagittal parameters lead to small sample subgroups. Further stratification e.g. due to peculiarity of the deformity of rib cage would cause even smaller subgroups, which significantly limits statistical analysis.
As you suggested, we discussed this interesting point in our limitations and we thank you for this advice making this research more interesting (page 15, Line 09-11).
Reviewer 2 Report
This is an interesting paper rich of data about in-brace correction of AIS. There is a need of similar studies, but there are some limits that reduce its helpfulness for readers.
The main limiti is that we don’t have post treatment data to compare with the in-brace and pre-treatment condition. I feel not so useful to have only in-brace correction. The best would be to have final results after treatment, and include also an objective monitoring of the compliance to show the changes of the radiographic parameters. In case final results are not available, at least short term results out of brace should be added.
About the analysis, I think it isn’t a good approach to make all these different analysis. I would run a regression using TK, SVA and pelvic parameters and all th others used. I would use these confounders as continuous variables instead of dicotomizing them
Author Response
Thank you for this direction. You raise a very critical point and we agree with your assessment that follow-up results after brace treatment would be very interesting.
The aim of this study was to investigate the immediate influence of brace on lumbar and thoracic spine in AIS patients and its possible influence on cervical spine. There is still lack of data on immediate in-brace therapy and its influence on the sagittal profile. Therefore, we aimed to address this gap. We believe that investigation of in-brace changes of the spinal profile sheds light on possible side effects of bracing which definitely should be the purpose of further studies.
During conceptualization of this study we discussed the necessity of follow-up measurements with our statistician and came to this conclusion:
The main part of this study population had an indication for brace therapy in 2018 and 2019. Therefore, a relevant part of study cohort is still undergoing brace treatment. At this time, follow-up data would contain a small study group. Stratification of the follow-up patients concerning sagittal profile (as used in this study) would not be possible. Furthermore, as already published in normal volunteers during the growth spurt, juvenile and adolescent spine underwent relevant sagittal profile changes of the lumbar and thoracic regions. This fact indicated potential changes of the sagittal profile also in AIS patients, independent if there was necessity of brace therapy or not. Consequently, we believe that follow-up data is needed for a large AIS study population post brace-therapy. This data has to be compared to data of follow-up of normal volunteers to reach enough statistical power. Thank you very much for this direction which should be explored in studies with larger sample size. We will address this issue in our prospective investigation, when we reach a follow-up cohort of 200-300 AIS patients with completed brace-treatment.
We discussed it in our limitations section (page 15, line 3-9). Thank you for this point!
As you suggested, regression analysis of TK and SVA and pelvic parameters is definitely a good option. About the analysis, we want to explain our thoughts by conceptualization of this study. With global analysis of whole study population we could show influence of in-brace-therapy on sagittal profile of lumbar, thoracic and cervical spine (first part in section results). But, due to heterogeneity of sagittal profile in AIS patients, we supposed different in-brace sagittal changes in dependency of pre-brace lumbar and thoracic profile. We discussed this with our statistician and we believe that this dichtomous data stratification gives a good opportunity to disentangle in-brace spinal profile changes in dependency of pre-brace lumbar normolordosis vs. hyperlordosis and thoracic normokyphosis vs. hypokyphosis. Furthermore, this stratification method for analyze of spinal profile in AIS patients was already introduced and published in former study in collaboration with Professor Virginie Lafage and Professor Frank Schwab [1].
We added the explanation for choosing this stratification in section methods and hope that you find it acceptable for possible publication process (page 3, line 32-34).
1. Akbar, M., et al., Sagittal alignment of the cervical spine in the setting of adolescent idiopathic scoliosis. J Neurosurg Spine, 2018. 29(5): p. 506-514.
Reviewer 3 Report
The authors provide an exhaustive data presentation of a well documented AIS patient' cohort. Though, the absence of a real long-term (years-long) follow-up as well as the lack of results after brace withdrawal removes the potential clinical interest of such description.
Futhermore, no important infos nor conclusions on c-spine parameters where described.
All in all, it seems a good departure point but needs further FU work and an adaptation of the objectives in order to turn this paper of significance.
Author Response
Thank you for your kind words about our paper. You raise a very critical point and we agree with your assessment that long-term follow-up results after brace treatment would be very interesting.
The aim of this study was to investigate the immediate influence of brace on lumbar and thoracic spine in AIS patients and its possible influence on cervical spine. There is still lack of data on immediate in-brace therapy and its influence on the sagittal profile. Therefore, we aimed to address this gap. We believe that investigation of in-brace changes of the spinal profile sheds light on possible side effects of bracing which definitely should be the purpose of further studies.
During conceptualization of this study we discussed the necessity of follow-up measurements with our statistician and came to this conclusion:
The main part of this study population had an indication for brace therapy in 2018 and 2019. Therefore, a relevant part of study cohort is still undergoing brace treatment. At this time, follow-up data would contain a small study group. Stratification of the follow-up patients concerning sagittal profile (as used in this study) would not be possible. Furthermore, as already published in normal volunteers during the growth spurt, juvenile and adolescent spine underwent relevant sagittal profile changes of the lumbar and thoracic regions. This fact indicated potential changes of the sagittal profile also in AIS patients, independent if there was necessity of brace therapy or not. Consequently, we believe that follow-up data is needed for a large AIS study population post brace-therapy. This data has to be compared to data of follow-up of normal volunteers to reach enough statistical power. Thank you very much for this direction which should be explored in studies with larger sample size. We will address this issue in our prospective investigation, when we reach a follow-up cohort of 200-300 AIS patients with completed brace-treatment Thank you for this point. We discussed it in our limitations section (page 15, line 3-9).
In terms of the cervical spine, we observed that brace treament leads to slight changes of cervical parameters of the lower spine (without changing of CL) and no changes of parameters of upper cervical spine. We discussed this phenomenon on page 13, line 4-14. Furthermore, we concluded that the influence of brace therapy on cervical spine is marginal and not existent in upper cervical spine (page 14, line 4-10).
We hope you find this discussion and conclusion about c-spine adequate and suitable for publication.
Round 2
Reviewer 3 Report
Thank you for your answers and content improvement.